# Nuclear Gene Transformation in the Dinoflagellate *Oxyrrhis marina*

**DOI:** 10.3390/microorganisms8010126

**Published:** 2020-01-16

**Authors:** Brittany N. Sprecher, Huan Zhang, Senjie Lin

**Affiliations:** Department of Marine Sciences, University of Connecticut, 1080 Shennecossett Rd, Groton, CT 06340, USA; brittany.sprecher@uconn.edu

**Keywords:** dinoflagellates, gene transformation, molecular biology, marine biology

## Abstract

The lack of a robust gene transformation tool that allows proper expression of foreign genes and functional testing for the vast number of nuclear genes in dinoflagellates has greatly hampered our understanding of the fundamental biology in this ecologically important and evolutionarily unique lineage of microeukaryotes. Here, we report the development of a dinoflagellate expression vector containing various DNA elements from phylogenetically separate dinoflagellate lineages, an electroporation protocol, and successful expression of introduced genes in an early branching dinoflagellate, *Oxyrrhis marina*. This protocol, involving the use of Lonza’s Nucleofector and a codon-optimized antibiotic resistance gene, has been successfully used to produce consistent results in several independent experiments for *O. marina*. It is anticipated that this protocol will be adaptable for other dinoflagellates and will allow characterization of many novel dinoflagellate genes.

## 1. Introduction

As widely distributed primary producers, essential coral endosymbionts, and the greatest contributors of harmful algal blooms and biotoxins in the ocean, dinoflagellates are a diverse group of unicellular protists with great ecological significance, evolutionary uniqueness, and numerous cytological and genomic peculiarities. While early diverging lineages share more similarities to typical eukaryotes, later diverging (namely “core”) dinoflagellates have immense and permanently condensed genomes with many chromosomes [1,2,3]; their genomes have a low protein–DNA ratio and histones are functionally replaced with dinoflagellate viral nuclear proteins (DVNPs) [4,5]; there are high numbers of repetitive non-coding regions and gene copies—in some species, up to ~5000 copies—organized in tandem arrays [6,7]; only 5–30% of their genes are transcriptionally regulated [7,8,9], with microRNAs appearing to be the major gene regulating mechanism [10]; and they have undergone extreme plastid evolution, transferring a massive quantity of plastid genes to the nucleus in most of the photoautotrophic species [11,12,13]. However, the molecular underpinnings of these unusual features remain elusive. In attempts to address the knowledge gap, an increasing amount of effort has been made in the last decade to analyze dinoflagellate transcriptomes [14,15,16,17,18,19,20,21,22,23,24,25,26,27] and genomes [10,28,29,30,31,32]. These experiments have provided not only extensive information on predicted genes and biological pathways, but also an even greater wealth of genes that have weak similarity to characterized proteins or no significant matches in databases. With the increasing volume of dinoflagellate transcriptomic and genomic data, the functional characterization of these novel genes has become a major bottleneck in translating system-level data into a mechanistic understanding of basic dinoflagellate biology, warranting the need for a dinoflagellate genetic transformation system.

Gene transformation attempts have been reported for dinoflagellates by four separate groups. Ten and Miller (1998) [33] utilized silicon carbide whiskers, polyethylene glycol (PEG), and vigorous shaking to introduce foreign DNA into *Amphidinium* sp. and *Symbiodinium microadriaticum* with a success rate of 5–24 per 10^7^ cells. Seventeen years later, Ortiz-Matamoros et al. used PEG, glass beads, shaking and, in some cases, co-incubation with the Gram-negative soil bacterium, *Agrobacterium tumefaciens,* to transform foreign DNA into *Fugacium kawagutii* (formerly *Symbiodinium kawagutii*)*, Symbiodinium microadriaticum*, and an unclassified Symbiodiniaceae species [34,35]. Neither of these reports used codon-optimized plasmids for dinoflagellate expression nor did the expression vectors used contain potential dinoflagellate promoters; moreover, both methods remain to be reproduced in other laboratories. 

Recently, studies have shifted to incorporate dinoflagellate DNA for gene delivery. Diao et al. (2018) report gene introduction into *Crypthecodinium cohnii*, by flanking an antibiotic gene with homologous arms of their dinoflagellate gene of interest, resulting in gene knockdown by homologous recombination [36]. While they reported successful knockdown of one gene, the study does not produce an expression system that could be adopted for other genes or dinoflagellates. In the most recent study, Nimmo et al. (2019) use plasmids containing dinoflagellate minicircle DNA with an antibiotic resistant gene and introduced the system successfully into the chloroplast genome of the dinoflagellate *Amphidinium carterae* [37]. Here, we report a successful nuclear gene transformation method for the dinoflagellate *Oxyrrhis marina.*

*O. marina* is a widespread and ecologically significant heterotrophic dinoflagellate. It is an established model species for both ecological and evolutionary research due to its easy cultivable nature, extensive studies related to feeding behavior and nutrition, and its position in dinoflagellate phylogeny [38,39,40,41,42]. As an early branching dinoflagellate, *O. marina* is not considered a “core” dinoflagellate [13,43,44,45]. Although it retains more typical eukaryotic features that are lacking in later diverging dinoflagellate taxa, it still shares many of the peculiar biological characteristics of the “core” dinoflagellates (e.g., trans-splicing, permanently condensed chromosomes, the use of DVNPs, fragmented mitochondrial genome, large genome) [16,44,45]; thus, it represents a good model for understanding dinoflagellate evolution [38,39,40,41,42,43,44,45,46]. In addition, *O. marina* has represented planktonic heterotrophs in experiments examining both how they feed and their nutritional value [47,48,49]. Through various studies as a prey species for copepods and rotifers, *O. marina* has been considered a trophic upgrade as they produce long-chain fatty acids, sterols, and essential amino acids that phytoplankton alone cannot [47,50,51]. Their nutritional value lead to the proposition of using *O. marina* as nutraceuticals for humans and agriculture [48]. 

Although *O. marina* lacks a published genome, several transcriptomic studies are available [16,52,53,54,55], and one interesting finding is that *O. marina* possesses a potential proton pumping rhodopsin with homology to proteorhodopsin [53,56]. Proteorhodopsin is a retinal protein/carotenoid complex that utilizes sunlight to pump protons across a membrane, a non-photosynthetic form of light harvesting [56]. Dinoflagellate species across the phylogenic tree have been found to possess proteorhodopsin homologs, allowing the translational study of this protein’s function in *O. marina* to the other dinoflagellate species [16,17,53]. Therefore, having a genetic transformation system in place for *O. marina* will improve our understanding of protist ecology, deepen our evolutionary understanding of dinoflagellates within their own branch and relative to other alveolates, allow exploration of the many predicted and novel dinoflagellate genes, and could tap into new industrial applications for *O. marina,* such as a food source or a potential alternative fuel. Additionally, since *O. marina* is a heterotrophic species, it will be easier to detect the expression of introduced florescent proteins without interference from chlorophyll florescence, as in photoautotrophic species. 

In this study, based on genomic and transcriptomic data from several dinoflagellates, we constructed a dinoflagellate expression system (named as DinoIII) that contains potential promoter and termination regions as well as important RNA elements. We incorporated a green fluorescent protein gene, *gfp*, (DinoIII-*gfp*) and a codon-optimized rifampin resistance gene (DinoIII-*arrO*) into DinoIII, and transformed this DNA as either PCR amplified fragments (excluding the plasmid bacterial backbone component for DinoIII-*arrO*), or as linear plasmid DNA (using a restriction enzyme to digest DinoIII-*gfp*) into *O. marina* using Lonza 4D-NucleofectorTM X system (Basel, Switzerland). The Nucleofector is a gene transformation system enabling the transfer of genes directly into the nuclei of the cells [57]. We have been able to repeat transformation for antibiotic resistance several times and verified the presence of both the antibiotic resistance gene and green fluorescent protein several months after transfection.

## 2. Methods

### 2.1. Culturing O. marina

*Oxyrrhis marina* CCMP 1795 was grown at 20 °C in autoclaved 0.22 mm filtered seawater (SW) on a 14:10 h light:dark cycle at a photon flux of ~100 μE m^−2^ s^−1^ and was fed *Dunaliella tertiolecta* CCMP1320 as prey [54]. Both species were purchased from the Provasoli-Guillard National Center of Marine Algae and Microbiota in West Boothby Harbor, ME, USA.

### 2.2. Testing O. marina Resistance to Selection Markers

Using 24-well plates, 1.5 mL of *Oxyrrhis marina* culture (~5 × 10^4^ cells/ mL) was added to 1.5 mL of SW already containing the desired selection marker (Table 1). The cultures were counted microscopically using a Sedgwick-Rafter counting chamber (Wildco, Yulee, FL, USA) for 7–10 days. Each selection marker tested was done in triplicate and a range of concentrations were tested in order to narrow down a concentration that resulted in total mortality within 7–10 days. Once the ideal concentration was found, it was experimentally repeated at least two to three times to validate the results.

### 2.3. Constructing Dinoflagellate Expression Vectors 

To optimize the utilization of our dinoflagellate expression system, several regions were amplified from dinoflagellate genomes and were incorporated to serve as the vector backbone. The first region (974 bp) comprises of DNA fragments including SL RNA, SRP RNA, several tRNAs, and U6 [58] from the dinoflagellate *Karenia brevis* and was named the DinoSL complex (Figure 1, Appendix A). This region was PCR amplified with DinoSL and KbrSRP-U6R1 primer set (sequences and Tm in Table 2) using the high fidelity PrimeSTAR HS DNA Polymerase (Takara, Kusatsu, Shiga, Japan) at 94 °C for 1 min, 30 cycles at 95 °C for 15 s, 58 °C for 30 s, and 72 °C for 1 min, and an additional elongation step at 72 °C for 10 min. The PCR product was run on 1% agarose to confirm the correct size, purified by passing through a DNA column (Zymo, Irvine, CA, USA), end-fixed, ligated into the pMD™19-T plasmid vector (Takara), and transformed chemically into *Escherichia coli* competent cells. Ampicillin was used to select for colonies harboring the region and plasmids were isolated and sequenced to identify the best clone, named pMD-Dino. 

From the *F. kawagutii* genome sequence data [10], we located the highly expressed light harvesting complex (LHC) gene. Its upstream “promoter” region (672 bp; Appendix A) and downstream “termination” region (812 bp; Appendix A) were PCR-amplified using the following primer sets: SymkaLHC5FN1 and SymLHC3_5R for the “promoter” and SymLHC5_3F and SymkaLHC3R1 for the “termination” region. All PCRs were performed at 94 °C for 1 min, 25 cycles at 95 °C for 15 s, 68 °C for 30 s, and 72 °C for 1 min, and 1 cycle of 72 °C for 10 min. The sizes of the amplicons were checked by electrophoresis and DNA was purified by passing through a DNA column (Zymo Research, Irvine, CA, USA). 

SymLHC3_5R and SymLHC5_3F were designed to contain an overhang of either a portion of the “termination” region or the “promoter” region in order to link the two PCR products; thus, the two products were used in an equal molar ratio as a template for the second PCR at 94 °C for 1 min, 5 cycles without primers at 95 °C for 15 s, 68 °C for 90 s, 20 cycles with SymkaLHC5FN1 and SymkaLHC3R1 at 95 °C for 15 s, 68 °C for 90 s, and an extra elongation step of 72 °C for 10 min. The single product was checked by electrophoresis to verify the amplicon size, gel isolated, and digested with *Spe*I and *Eco*RI as SymkaLHC5FN1 had a *Spe*I site and SymkaLHC3R1 had an *Eco*RI site added to their 5′ ends for easy incorporation into pMD-Dino harboring the SL RNA, SRP RNA, several tRNAs, and U6 region. The pMD-Dino vector was digested with *Xba*I and *Eco*RI and treated with alkaline phosphatase to avoid self-ligation. After 3.5 h of digestion both products were purified by ethanol precipitation and ligated overnight in a 2:1 molar ratio (LHC product:vector) and transformed into competent *E. coli* cells. The colonies obtained were picked randomly, and plasmids were isolated and sequenced to identify the best clone harboring the correct DinoSL Complex-LHC region sequence, giving rise to the dinoflagellate expression vector backbone, DinoIII (5137 bp; Figure 1, Appendix A).

SymLHC3_5R and SymLHC5_3F primers were designed to also have an *Xba*I and *Bgl*II site in between the “promoter” and “termination” regions so that a gene, either a reporter or an antibiotic resistant gene, could be inserted in the correct orientation. Accordingly, both a *gfp* gene and a rifampin resistance gene were incorporated into the expression vector, to yield DinoIII-*gfp* and DinoIII-*arrO*, respectively (Appendix A, Appendix A). For DinoIII-*gfp*, the crystal jelly *Aequorea victoria gfp* was amplified from the pGlo™ Plasmid (Bio-Rad, Hercules, CA, USA; GenBank accession # U62637) using gfpNF2 and gfpNR at 94 °C for 1 min, 5 cycles at 95 °C for 15 s, 55 °C for 30 s, and 72 °C for 30 s, 25 cycles at 95 °C for 15 s, 62 °C for 30 s, and 72 °C for 30 s, and an extra elongation step of 72 °C for 10 min. 

For *arrO*, a homolog to rifampin ADP-ribosylating transferase from bacterium *Citrobacter freundii* was found on GenBank (accession # NC_019991) and was codon-optimized (Appendix A) for *O. marina* based on codon usage data from reported *O. marina* genes available on GenBank, and was synthesized through GeneArt (Thermo Fisher Scientific, Waltham, MA, USA). Upon arrival the synthesized *arrO* was PCR amplified at 94 °C for 1 min, 30 cycles at 95 °C for 15 s, 57 °C for 30 s, and 72 °C for 30 s, and an extra elongation step of 72 °C for 10 min with arr2F and arr2R primers. Both the *gfp* and *arrO* genes had a *Spe*I site at the 5′-end and a *Bcl*I site at the 3′-end; thus, after their PCR amplification, the single products were checked on a gel, passed through DNA columns to purify, and digested with *Spe*I and *Bcl*I for 3.5 h. At the same time, DinoIII was digested with *Xba*I and *Bgl*II and treated with alkaline phosphates. After digestion, the *gfp* and *arrO* genes were ligated into DinoIII overnight in a 2:1 molar ratio and were transformed into competent *E. coli* cells. Plasmids were isolated and sequenced to identify the clones containing correct sequences of DinoIII-*gfp* and DinoIII-*arrO* (Appendix A, Appendix A). 

### 2.4. Optimizing Promoter Region

To optimize the expression of transformed genes for *O. marina,* we set out to find a promoter region for their highly expressed proteorhodopsin genes (2–4 × 10^6^ copies/ng total RNA—twice that of mitochondrial *cox1*) [54]. To do this we used the OxyRhodF2 and OxyrhodR primer set [54], under thermal cycle conditions with an extended extension time to favor long amplicons that cover two or more tandem repeats of the gene: 94 °C for 1 min, 25 cycles at 95 °C for 15 s, 60 °C for 30 s, and 72 °C for 90 s, and an extra elongation step of 72 °C for 10 min. Bands of ~600 and ~1200 bp were gel purified, cloned, and sequenced to identify a potential promoter. This yielded an intergenic region between rhodopsin tandem repeats, with the following g/t rich sequence: aattttgggagttgggctggaagatggggttggtggggatcgggggagaggtgactggtgtgtggtcgag. We added this sequence to the 5′-end of the *arrO* gene through GeneArt (Thermo Fisher Scientific, Waltham, MA, USA) and incorporated this *arrO*-N sequence into the DinoIII vector (DinoIII-*arrO*-N; Appendix A, Appendix A) as described above.

### 2.5. Introducing DNA into O. marina Using Lonza’s Nucleofector

*O. marina* cultures were fed with *D. tertiolecta* three days before transformation in order to reach high cell densities and given enough time to clear all the *D. tertiolecta* cells from the culture. Taking advantage of the species photo-tactic behavior, *O. marina* cells were concentrated using a flashlight, allowing cells to swim toward the light, consequently gathering only healthy cells from the culture. The cultures cell numbers were counted microscopically using a Sedgwick-Rafter counting chamber. 

Electroporation was carried out using Lonza 4D-Nucleofector™ X Unit system in 16-well Nucleocuvette™ Strips using the manufacturer’s SG and Supplemental 1 solutions. DinoIII-*gfp* and DinoIII were digested with *Eco*RI and introduced as linear plasmid and DinoIII-*arrO*, DinoIII-*arrO*-N, DinoIII plasmids were PCR amplified, to produce linear fragments that only contained the dinoflagellate DNA portion. DinoIII linear plasmid DNA or PCR amplified linear fragments were used as pulse controls (PC). PCR was carried out using Illu-DSL and SymkaLHC3R1 primers at 94 °C for 1 min, 25 cycles at 95 °C for 15 s, 68 °C for 90 s, and an extra elongation step of 72 °C for 10 min. The linear plasmid and PCR product were checked through electrophoresis, retrieved, and concentrated to 1μg/μL using a Millipore Microcon DNA Fast Flow Column (Burlington, USA). For each transformation well, 16.4 μL of solution SG, 3.6 μL of Supplemental 1 solution, and 2 μL of PCR product/linear plasmid DNA were used as transformation solution.

For every well, ca. 2.5 × 10^5^ cells were added and all the cells for the experiment (including the controls) were collected in 50 mL tubes. The cells were centrifuged at 2500× *g* for 3 min, enough to form a pellet at the bottom of the tubes, and all but ~2–3 mL of medium was removed. The cells were transferred into 1.5 mL tubes, centrifuged at 900× *g* for 1 min, and all remaining liquid was removed. The cells were re-suspended in the transformation solution and 22 μL was added to each well. After an initial optimization test, the following electroporation settings were used for further experiments: DS-137, DS-130, DS-138, DS-134, DS-150, ED-150, DS-120, no pulse controls (NPCs), and all of the mentioned pulse code settings with just linear PCR DinoIII fragments or linear *Eco*RI digested DinoIII plasmid DNA (PC)**.** For every experimental trail, NPCs were included in triplicate and PCs were also included, mirroring the experimental treatment design.

Immediately after electroporation, 80 μL of the same seawater medium (SW) that *O. marina* was cultured on, but with an antibiotic cocktail, AKS (100 μg/mL ampicillin, 50 μg/mL kanamycin, and 50 μg/mL streptomycin), was added to each well. All of the volume was gently transferred into 24-well plates where each well already contained 1.4 mL of the same SW+AKS medium. The transformed cells were allowed to recover for three days. For the DinoIII-*gfp* transformations, cells were examined microscopically under blue light for *gfp* expression. When bright GFP was visualized, 80 μL of the culture was mixed with 2.4 μL of 0.5 M KOH (final volume ~15 mM KOH) in order to slow the cell movement for video recording. For DinoIII-*arrO* and DinoIII-*arrO*-N cells, 750 μL (half the well volume) were transferred to new 24-well plates and 750 μL SW+AKS containing 450 μg/mL of rifampin was added to all wells in both plates, so that the final concentration of rifampin was 225 μg/mL. On the third day under antibiotic selection, 5 μL of *D. tertiolecta* culture in 225 μg/mL rifampin medium was added to each well. New antibiotic solution was added every 3 weeks but the concentration was dropped down to 200 μg/mL once all NPC and PC conditions had died off to allow for greater cell growth and *D. tertiolecta* in 225 μg/mL rifampin medium was supplied in small concentrations (5 μL/well) whenever they were no longer detected in the medium. For DinoIII-*arrO*-N cultures, the rifampin concentration was kept at 225 μg/mL due to their ability to grow to higher cell concentrations.

### 2.6. Detecting the Transformed Gene and Its Expression

Total DNA was isolated from both wild-type (WT) and +*arrO/*+*arrO*-N *O. marina* cultures using our CTAB method [59] and total RNA was isolated using the Trizol-Chloroform method in combination with Zymo Quick RNA Miniprep Kit (Irvine, CA, USA) [60]. As the transformed cultures grew slowly under antibiotic pressure, only 200–400 cells were available. These cells were divided into two and collected on TSTP Isopore 3 μm membrane filters (MilliporeSigma, Burlington, MA, USA) and DNA and RNA was isolated.

Several first-strand cDNA preparations were made with RNA isolated using ImProm-II™ Reverse Transcriptase (Promega, Madison, WI, USA) following the manufacture’s protocol with random hexamer (N6), oligo(dT)_18_ (OdT), and modified OdT (MdT; Table 2) as the primers. If cell numbers were very low, only OdT was used for cDNA synthesis to maximize cDNA production. Negative controls were included where no reverse transcriptase was added and was instead replaced with diethyl pyrocarbonate (DEPC) water.

PCR was performed using both the DNA and cDNA as templates with primer set arr2Q1Fa-arr2Q1Ra. Due to the low number of transformed cells used in DNA and RNA isolation, this PCR did not yield detectable amounts of products. The PCR products were then diluted 1,000- and 10,000-fold and used as a template for a nested PCR with arr2Q1F-arr2Q1R as the primers (Table 2) for quantitative PCR. The products were run on a gel and sequenced directly to ensure the *arrO* gene was correctly amplified.

### 2.7. Transforming D. tertiolecta with DinoIII Using Lonza’s Nucleofector

To investigate whether our transformation protocol could transform *D. tertiolecta* cells, which would potentially give fluorescence or antibiotic resistance to *O. marina* that fed on this alga, *D. tertiolecta* cells were transformed with linear DinoIII-*gfp* plasmid DNA and PCR amplified fragments of both *arrO* and *arrO*-N, following the exact same methods described above for *O. marina.* Every pulse code was done in triplicate for each of the different plasmids. *D. tertiolecta* cells with DinoIII-*gfp* plasmid DNA were examined microscopically under blue light for *gfp* expression 24 h post-transformation and the entire volume was fed to wild-type *O. marina* cells. The *O. marina* cells were examined microscopically under blue light for green fluorescence for several weeks.

After the 3 day recovery phase, *D. tertiolecta* cells transformed with either *arrO* or *arrO*-N were divided into two. One-half (750 μL) was put under different antibiotic concentrations (0, 225, and 300 μg/mL rifampin) and the other was fed to wild-type *O. marina* cells in 225 μg/mL rifampin. Both *D. tertiolecta* and *O. marina* were counted microscopically using a Sedgwick-Rafter counting chamber for 7–14 days, the amount of time necessary to kill off wild-type *O. marina* cells.

## 3. Results

### 3.1. Construction of Dinoflagellate Backbone Expression Vector

Initially, a DNA fragment with a cluster of small RNA genes (including SL RNA, SRP RNA, several tRNAs, and U6 snRNA) was amplified from the dinoflagellate *Karenia brevis* (GenBank accession # FJ434727) to form the “DinoSL Complex” region (Appendix A) and was inserted into the pMD™19-T plasmid vector (Takara, Japan) to serve as the vector’s skeleton for a series of modifications. After the addition of more dinoflagellate elements, the upstream (Appendix A) and downstream regions (Appendix A) of *F. kawagutii’s* highly expressed light harvesting complex gene, a functional dinoflagellate backbone vector was achieved, named DinoIII (5137bp; Figure 1; Appendix A).

### 3.2. Transformation using Lonza’s 4D-Nucleofector™ X Unit System

We went through an extensive cell optimization protocol for *O. marina* using the 4D-Nucleofector™ X Unit (Lonza Bioscience, Basel, Switzerland), an electroporator intended to work on cells that are challenging to transform, and identified seven adequate pulse code settings, the exact electroporation conditions are proprietary. We used these seven pulse codes for follow-up experiments (Table 3). Each pulse code had varying levels of success; for some *gfp* expression was observed in several cells, whereas others had cultures grow to large enough populations under antibiotic pressure to allow for RNA and DNA isolation. Taking all the data into consideration, the pulse codes that showed overall strongest performance were DS-134, DS-137, and DS-120. Nevertheless, we recommend the use of all seven settings in the first optimization tests for this species. For other algae to be studied, full optimization tests with the other available solutions should be utilized when using the 4D-Nucleofector™ X Unit.

### 3.3. GFP Expression

The reporter expression vector, DinoIII-*gfp* (Appendix A, Appendix A), was introduced as linear DNA to *O. marina* cells and the presence of fluorescence was examined microscopically from the third day on. For several weeks we only observed a very dim green signal, but after three months the brightness of the green signal markedly increased for two transformed pulse code settings, DS-137 and DS-130. We discovered that even natural *O. marina* cells would give dim green-yellowish autofluorescence under blue light when fixed with Paraformaldehyde or other commonly used fixatives, making it very challenging to take a clear image of the *gfp* expressed cells, a necessity to detect the exact location of GFP in the cell. We endeavored to take clear microscopic videos of *O. marina* cells that expressed *gfp*, but with limited success due to the swift movement of the cells under the microscope. After many trials, we found that adding KOH (at 15 mM final concentration) could slow down the movement of *O. marina* just enough for us to take a relatively clear microscopic video under 400× magnification (Appendix A), although cells would burst in approximately 5–10 min. Out of ~30 trials, the expressed GFP signal was particularly strong in three and weaker in others, and in each trial approximately less than 0.1% of the total *O. marina* population showed detectable GFP signal from healthy swimming cells, making the isolation of the transformed cell line challenging without a selection marker.

### 3.4. Rifampin Resistance as a Selection Marker

To facilitate selection of transformed *O. marina* cells, we screened commonly used antibiotics and other selective agents (Table 1). The growth inhibiting effects of the selection agents on wild-type *O. marina* varied widely, with only three resulting in lethal concentrations. Rifampin turned out to be the most effective for *O. marina*, as demonstrated in the growth curves for wild-type cultures treated with this antibiotic (Appendix A).

Rifampin is an antibiotic used to treat tuberculosis, leprosy, and Legionnaire’s disease and its resistance in bacteria is due to rifampin ADP-ribosyltransferase activity [61]. Rifampin has been used to treat several *Plasmodium* species [62,63] and acts on their apicoplast by blocking transcription. The exact mechanism of how rifampin works on *O. marina* is unknown, and the uncovering of this mechanism is beyond the scope of this study; however, *O. marina* has also been found to have plastid gene remnants [64], which could be a potential target of this antibiotic, or *O. marina* sensitivity could be due to an unknown off-target effect. We found a bacterial rifampin resistance gene, rifampin ADP-ribosylating transferase, in GenBank and to facilitate its expression in *O. marina*, we optimized the codons based on *O. marina’s* codon preference (Appendix A). We then synthesized and introduced this codon-optimized gene, *arrO,* into *O. marina* through our DinoIII vector (DinoIII-*arrO*; Appendix A, Appendix A) and achieved expression of *arrO*, which was verified in several ways.

First, the transformed cell culture survived and grew while the wild-type, including pulse controls and no pulse controls, died completely in rifampin-containing medium. Second, after approximately one month we isolated RNA and DNA from both the transformed cells cultured in rifampin-containing medium and a wild-type culture grown in rifampin-free growth medium and performed reverse-transcription PCR. Using nested PCR, we detected *arrO* only in the experimental treatment and not in the wild-type (Figure 2). In addition, we sequenced the PCR product and confirmed that it was *arrO*. Finally, the expression of *arrO* was detected from the cDNA synthesized using Oligo-dT as the primer (Figure 2A), indicating the transcript of *arrO* was polyadenylated, a phenomenon best known for occurring mostly in eukaryotes mRNA.

Although the cells survived for more than one month, the population increased slowly and initially did not seem healthy, probably due to low expression efficiency of the resistance gene. We attempted to increase the expression efficiency of our DinoIII vector by incorporating the intergenic region between *O. marina* rhodopsin tandem repeats, a potential promoter for this highly expressed protein [54]. After introducing the PCR fragment of the amended vector, DinoIII-*arrO-*N (Appendix A; Appendix A), into *O. marina* cells, the cultures had an approximately 5 to 10-fold increase in growth rate under antibiotic selection and were able to maintain populations for longer periods of time. The expression of *arrO* increased 9-fold compared to the original DinoIII-*arrO* vector when examined three months after transfection (Figure 3B). We still, when writing this manuscript, have a DinoIII-*arrO*-N cell line in culture (+a*rrO*-N and wild-type cell survival under rifampin are in Appendix A). Note, both NPC and PC conditions were found to die within 7 to 14 days.

### 3.5. Testing the Potential Interference of Dunaliella tertiolecta in O. marina Transformation

To examine the possibility that our observed *O. marina* transformation was an artifact due to transformation of the prey, *D. tertiolecta*, the same protocols described above were applied to *D. tertiolecta* cells. The *D. tertiolecta* cells transformed with linear DinoIII-*gfp* plasmid DNA were examined microscopically 24 h later and were observed for green fluorescence. We then fed the transformed *D. tertiolecta* cells to wild-type *O. marina* cells. These *O. marina* cells never exhibited the green fluorescence visualized from the originally transformed *O. marina* cells (Appendix A) when monitored microscopically for several weeks.

We also transformed *D. tertiolecta* cells with PCR amplified fragments from DinoIII-*arrO*-N and DinoIII-*arrO*, and the transformed *D. tertiolecta* cells were either allowed to grow in antibiotic medium or were fed to wild-type *O. marina* cells. *D. tertiolecta* cells did show some growth inhibition by the antibiotics (Figure 4A,B), but the “transformed” *D. tertiolecta* culture did not grow better than the control cells, and did not exhibit dose-dependent growth inhibition by the antibiotics; thus, indicating introduction and expression of DinoIII-*arrO*-N and DinoIII-*arrO* in the prey alga was not successful. Additionally, when the “transformed” *D. tertiolecta* cells were fed to wild-type *O. marina* cells and placed under antibiotic selection, the wild-type *O. marina* cells died in 7–15 days, similar to the NPC and PC conditions mentioned above, while the transformed *O. marina* cells that were fed wild-type *D. tertiolecta* cells maintained a population (Figure 5 and Appendix A).

## 4. Discussion

In order to improve understanding of basic dinoflagellate biology, a gene transformation protocol is urgently needed to characterize the function of dinoflagellate genes, particularly the vast number of nuclear genes, the majority of which have weak or no match to known genes. A robust and reproducible protocol has been long awaited. After testing multiple methods (including previously reported ones) and numerous conditions, we have found a passage and herein report a genome-targeted transformation method using a dinoflagellate *gfp* vector (DinoIII-*gfp*) and two dinoflagellate rifampin resistance vectors (DinoIII-*arrO* and DinoIII-*arrO-*N) that were developed based on dinoflagellate genomic and transcriptomic data.

Our efforts began with utilizing expression vectors from the previous reported dinoflagellate transformations [34,35]. This expression system utilized the plant CaMV 35S and *nos* promoters to drive expression of plasmids [34,35]. The *nos* and CaMV 35S promoters have been used extensively for plant transgenic studies, and the CaMV 35S promoter functions in both bacteria and animal systems [65] as well. These vectors contained the herbicide resistant gene, Basta (glufosinate), as well as several *gfp* fusion genes. Because *O. marina* is not sensitive to Basta, we were looking for green fluorescence as indication of successful transformation, but it was not observed. Therefore, we developed a series of dinoflagellate expression vectors based on existing dinoflagellate transcriptomic and genomic data, mirroring what was previously done for the two model alveolates, *Plasmodium falciparum* and *Tetrahymena thermophila* [66,67].

In attempts to construct a vector that could drive expression of any gene in any dinoflagellate species, we included dinoflagellate sequences from two different, phylogenetically separated species, *F. kawagutii* and *K. brevis.* Although both are “core” dinoflagellates, *F. kawagutii* belongs to the typical dinoflagellate group that fixes CO_2_ with form II Rubisco and contains a secondary endosymbiosis plastid with peridinin as the dominant accessory, while *K. brevis* instead utilizes form I Rubisco and contains a tertiary replacement plastid with fucoxanthin as the dominant accessory pigment [68]. The combination of the included elements (potential promoter, terminator, and RNA elements) was able to drive the expression, albeit at a low level, of the inserted genes in *O. marina*.

In order to increase the expression level, we identified a G-rich intergenic region between the highly expressed rhodopsin genes, and incorporated it into our DinoIII-*arrO* vector, yielding visually higher cell survival under antibiotic pressure, indicative of stronger expression of the rifampin resistance gene. When comparing our intergenic region to the intergenic regions between luciferase tandem repeats, a region that has been suspected to be a promoter [69], our sequence is much shorter, only 70 base pairs compared to ~200–2000, and has no real sequence matches in public databanks. No proven promoter exists for dinoflagellates at this time and it is uncertain if the additional sequence contains a promoter. Previous research looking at the binding affinity of *Crypthecodinium cohnii* TATA-binding protein (TBP) homolog and *F. kawagutii* genomic content suggests that dinoflagellates have replaced the typical eukaryotic TATA box with a TTTT motif [10,70], which is present and begins at position −68 in our intergenic region and is also present in our “promoter” region, with the first motif beginning at position −140. Whether or not these sequences are important can be evaluated in future studies using our method.

*O. marina* is a naked dinoflagellate that had a difficult time withstanding the physical forces used in previously reported dinoflagellate transformation methods. Electroporation allows DNA to pass through temporary pores in an organism’s membrane and has been utilized in many organisms but requires the removal of seawater and replacement with electroporation buffers [71]. A new electroporation model, Lonza’s 4D-Nucleofector, provides a user with a score of built-in pulse settings and solutions that remove salts but help maintain dinoflagellates osmolality. The machine has been designed for rapid optimization of both buffer and electric pulse conditions, allowing delivery of nucleic acid substrates into the nucleus [57]. The Nucleofector has been widely used on a variety of organisms and cell types and has recently been successfully used on two difficult to transfect marine protists, choanoflagellates and diplonemids [72,73]. For *O. marina,* seven pulse code settings (Table 3) and one solution, SG, allowed the expression of genes in DinoIII vectors. No one pulse code performed the highest across our different criteria. It is interesting to note that DS-120 yielded the highest cell number and DS-137 the lowest when examined 24 h post-transformation. Unfortunately, these settings are proprietary, and no correlation of the pulse settings can be extracted.

We were successful in taking microscopic videos of *O. marina* cells expressing the introduced *gfp* but were unable to obtain still images necessary for visualizing the exact subcellular location of the expressed protein. From the videos, it appears that GFP was concentrated in a small area rather than diffusely distributed as usually expected. Because GFP is a small-sized protein, it has been found to enter the pores of the nucleus in animal, plant, and yeast cells and can be found in either the cytoplasm or nucleus rather than a definite location [74]. Based on observations of the live cultures, only about several hundred to several thousand cells survived the transformation treatment; among them, less than 0.1% of the population actually showed *gfp* expression. The green fluorescence signal was dim initially, but the brightness markedly increased after three months for two pulse code settings (Appendix A). Without a selection agent, the percentage of *O. marina* cells expressing *gfp* decreased over time as the non-fluorescing cell population increased and the green signal became no longer visible after 4 months. The reproducibility for obtaining the same level of GFP expression and the isolation of the GFP-expressing cells has been incredibly challenging due to the lack of selection for the transformed cells and is the reason why we subsequently focused on antibiotic resistance.

Availability of an appropriate selection marker is crucial for yielding a useful transformed cell line. After extensive testing, rifampin was found to be effective for *O. marina*. Rifampin is a very strongly pigmented antibiotic that appears to be very light sensitive. Due to this characteristic it is important to keep *O. marina* in lower light settings when under antibiotic selection, keep the cultures fed, and continue to add new antibiotic medium to the transformed cell lines. For different strains of *O. marina*, it is important to test each strain first to determine optimal antibiotic concentrations that can be used to select transformed cell lines.

Ideally, a selection marker and a reporter gene can be located on the same plasmid, allowing for dual expression. We attempted to put both the *arrO* and *gfp* genes within one single DinoIII vector in multiple arrangements (with or without stop codon in between, fused or not fused) but, unfortunately, the simultaneous expression of both genes was not achieved. Future studies using *arrO* and *gfp* with the rhodopsin intergenic region in between could potentially get over this hurdle. We also attempted to amplify DinoSL-containing *arrO* from transformed *O. marina* cDNA libraries but were unsuccessful. This is probably because the transformed DinoIII-*arrO/arrO*-N is incorporated into a long 5′-noncoding region environment, making it difficult to amplify the long DinoSL-containing *arrO*/*arrO*-N sequence. Alternatively, this could be because the introduced gene has not yet been completely incorporated into the gene expression system of *O. marina* within the experimental period, which is extremely short from an evolutionary perspective.

To exclude the possibility that the transformation we observed (both GFP fluorescence and the rifampin resistance) was due to interference of the prey alga, *D. tertiolecta*, we conducted control transformation runs for *D. tertiolecta* and fed these cells to wild-type *O. marina*. These results clearly rule out the possibility and bolster our success in *O. marina* transformation.

Because *O. marina* is an early diverging dinoflagellate and not considered a “core” dinoflagellate, there are several important molecular differences between *O. marina* and the other typical dinoflagellates. For instance, mitotic cell division is driven by intranuclear spindle rather than extranuclear spindle [75,76], mRNA editing of mitochondrial genes is not found in *O. marina* [46] but exists in other dinoflagellates, and *O. marina* is thought to have fewer gene copies [77] (5–33 copies based on limited data currently available) when compared to “core” dinoflagellates (up to several thousand [2]). These differences may have made nuclear transformation easier for *O. marina* when compared to other dinoflagellates; however, using the same backbone vector (DinoIII) and similar transformation technique, we have also succeeded in transforming one of the “core” dinoflagellate species, *Karlodinium veneficum* (under revision). Furthermore, the DNA elements included in the expression vector were mostly derived from the “core” dinoflagellate species Symbiodiniaceae and *Karenia*, which belong to two distinct lineages. The successful expression in *O. marina* of the introduced genes with these elements is a promising sign of adaptability of our protocol for transforming “core” dinoflagellate species. In addition, the success on *O. marina* may also lend a prototype to transformation efforts on dinoflagellate-related organisms such as *Perkinsus* and other alveolates.

## 5. Conclusions

Despite the proven challenges, we have developed a dinoflagellate expression system and successfully used it to express foreign genes in the dinoflagellate *Oxyrrhis marina*. The success is confirmed by detecting expression of the introduced genes, including GFP and antibiotic resistance genes and excluding possibilities of artifacts using various controls. The gene transformation tool developed here makes the species an even more valuable model, as *O. marina* has been extensively studied as a model species for heterotrophic protists and dinoflagellates and is easy to cultivate. As an early diverging lineage of dinoflagellate, having a genetic transformation system in place for *O. marina* will allow a deeper understanding of basic dinoflagellate biology. With various promoter elements from core dinoflagellate species, the dinoflagellate backbone vector developed has the potential to work across the dinoflagellate phylogenetic tree, and the transformation protocol reported here will prove useful for transforming other dinoflagellate and related alveolate species.

## Figures and Tables

**Figure 1 microorganisms-08-00126-f001:**
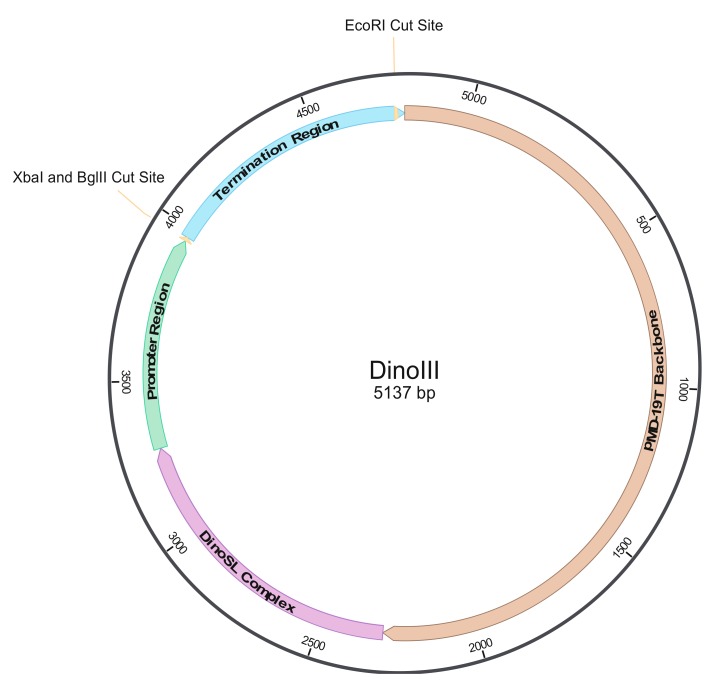
Structure of the dinoflagellate expression vector, DinoIII. The bacterial pMD-19TM T-Vector portion is shown in brown, while other colors depict dinoflagellate elements, including the DinoSL Complex region (purple; containing *Karenia brevis* SL RNA, SRP RNA, several tRNAs, and U6 region) and the Promotor (light green) and Termination (light blue) Regions, which are the upstream and downstream regions of the *Fugacium kawagutii* light harvesting complex, respectively. *Xba*I and *Bgl*II cut sites, depicted with a line, allow for easy gene incorporation in the proper orientation. *Eco*RI site depicts where DinoIII/DinoIII-*gfp* is digested to form linear plasmid DNA.

**Figure 2 microorganisms-08-00126-f002:**
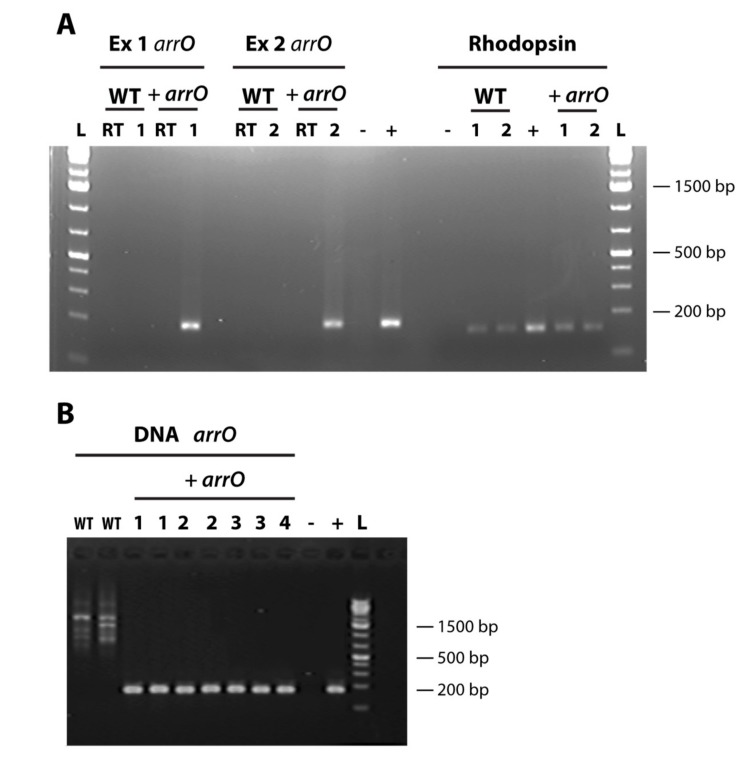
Detection of rhodopsin and *arr*O (codon-optimized rifampin-resistant gene) in transformed and wild-type *Oxyrrhis marina* cells; (**A**) *arr*O gene detected in experimental cDNA OdT libraries through a nested PCR in two separate transformation events (Ex 1 and Ex 2) and *O. marina’s* rhodopsin gene detected in all experimental cDNA libraries (+ *arr*O) and wild-type (WT) libraries; “RT”, no reverse transcriptase control; “L”, GeneRuler™ 1 kb Plus DNA Ladder (Thermo Scientific); “-”, no template control; “+”, plasmid positive control for *arr*O and *O. marina* gDNA for rhodopsin. (**B**) *arrO* gene detected in DNAs of transformed *O. marina* (in experiments 1-4) through a nested PCR; “L”, GeneRuler™ 1kb Plus DNA Ladder (Thermo Scientific); “WT”, wild-type; “1, 2, 3, 4”, samples from experiment 1, 2, 3, 4, respectively; “+ *arrO*”, transformed cells; “-”, no template control; “+”, plasmid positive control.

**Figure 3 microorganisms-08-00126-f003:**
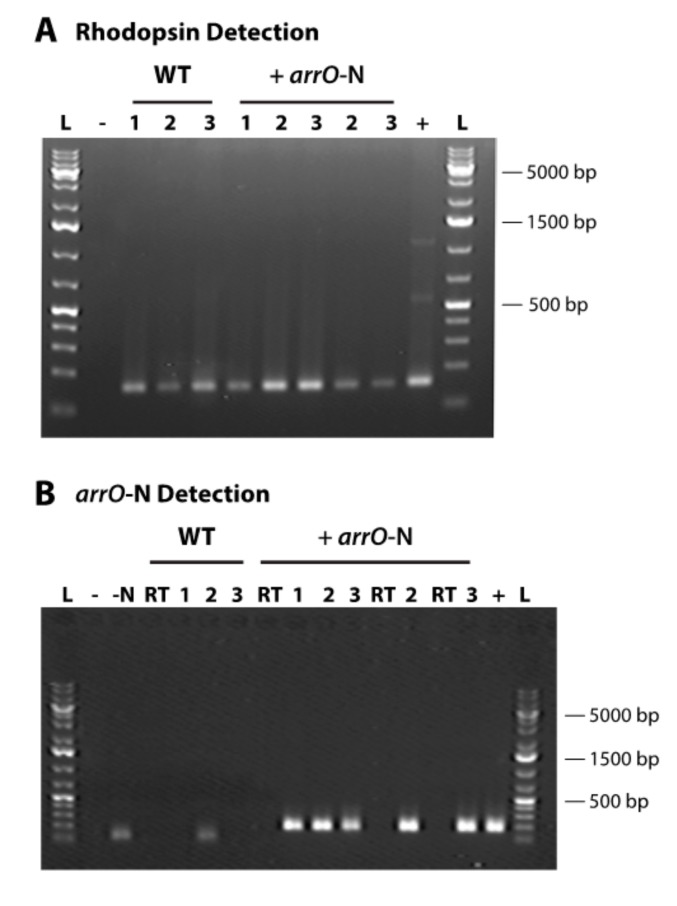
Detection of rhodopsin and *arrO*-N (*arrO* plus rhodopsin intergenic region) gene in three different transformation events and in wild-type *Oxyrrhis marina* cells; (**A**) *O. marina’s* rhodopsin gene detected in cDNA libraries for both transformed (+ *arrO*-N) and wild-type cells (WT); (**B**) *arrO*-N gene expression detected in the cDNAs of transformed *O. marina* through nested PCR; “L”, GeneRuler^TM^ 1kb Plus DNA Ladder (Thermo Scientific); “WT”, wild-type; “+ *arrO*-N”, transformed cells; “-”, no template control in first PCR; “-N”, no template control in nested PCR; “RT”, no reverse transcriptase control; “+”, plasmid positive control for *arrO*-N gene and *O. marina* gDNA for rhodopsin gene; “1”, N6 library; “2”, OdT; “3”, MdT library.

**Figure 4 microorganisms-08-00126-f004:**
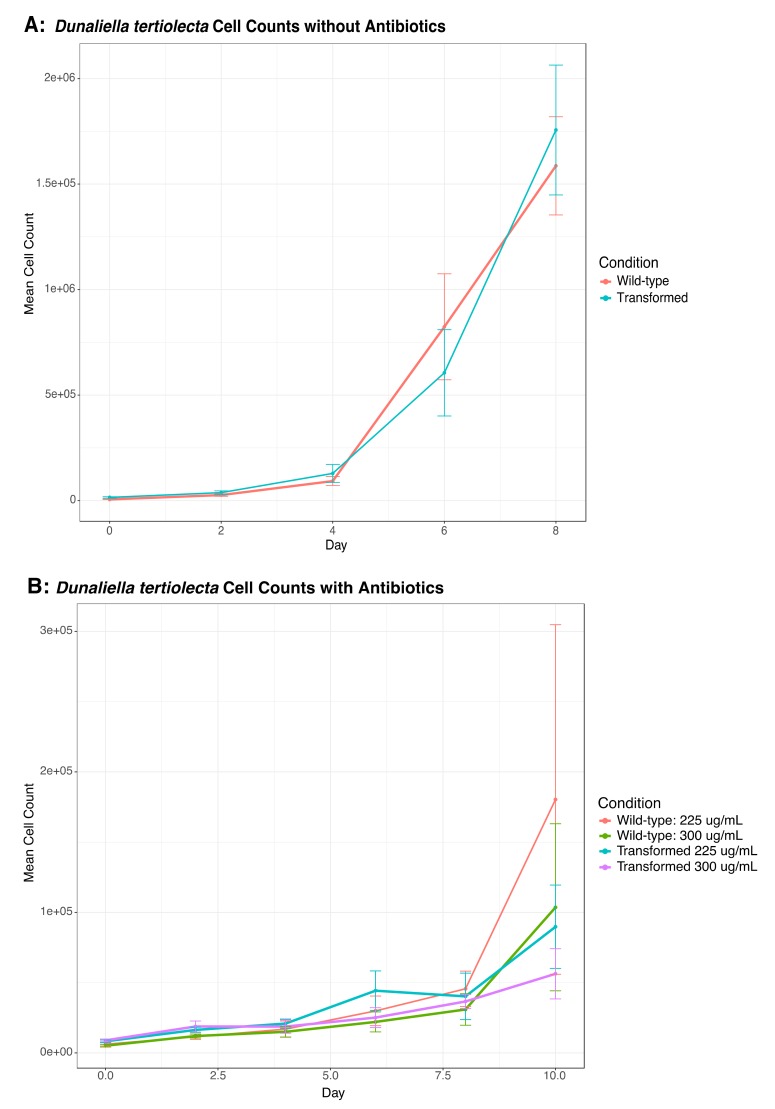
Cell counts for *D. tertiolecta* cells that were either transformed or wild-type cells; (**A**) Both transformed and wild-type growth curves for *D. tertiolecta* without antibiotics; (**B**) *D. tertiolecta* cells grown under rifampin at either 225 or 300 μg/mL. The center values represent the means, with standard deviation as error bars. The results indicate that antibiotic resistance was not conferred when exposing *D. tertiolecta* to *O. marina’s* transformation procedure.

**Figure 5 microorganisms-08-00126-f005:**
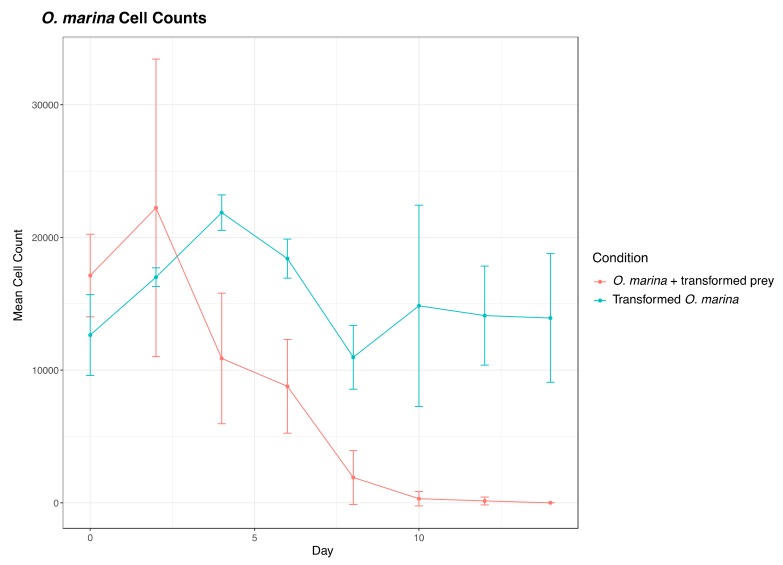
Comparison of *O. marina* cell counts under 225 μg/mL rifampin between wild-type *O. marina* fed transformed *D. tertiolecta* (“*O. marina* + transformed prey”) and transformed *O. marina* cells (*O. marina* +*arrO*-N) fed wild-type *D. tertiolecta*. The marked difference indicates that our observed *O. marina* transformation was not an artifact from *D. tertiolecta* transformation. The center values represent the means, with standard deviation as error bars from results of different pulse codes. Note that transformed *O. marina* stopped growing after 10 days because no additional prey was added to the culture since day 0.

**Table 1 microorganisms-08-00126-t001:** Different selection markers used on *O. marina*. The reported values are in μg/mL and are the amount of selective agent required to kill the entire population in 7–10 days. The antibiotics found to not work are noted by “n/a” and followed by the highest concentrations tested in parenthesis.

	Amphotericin B	Ampicillin	Basta	Blasticidin	Chloramphenicol	Formaldehyde	Geneticin (G418)	Hygromycin B	Kanamycin	Paromomycin	Rifampin	Streptomycin
*Oxyrrhis marina*	n/a (1000)	n/a (1000)	n/a (500)	n/a (500)	50	40	n/a (1000)	n/a (2000)	n/a (2000)	n/a (1500)	225	n/a (1000)

**Table 2 microorganisms-08-00126-t002:** Primers used in the present study.

Primer Name	Sequence Information	Tm	Polymerase Used
SymkaLHC5FN1	GAGAACTAGTAAGTCCCGTGGCTGTCATATCTAG	68 °C	Takara PrimeSTAR HS DNA Polymerase
SymLHC3_5R	GACTCCTGGCCGAGATCTTCTAGAGGCTCCGAAATTTGGTCTAAGCAC	68 °C	Takara PrimeSTAR HS DNA Polymerase
SymLHC5_3F	CCAAATTTCGGAGCCTCTAGAAGATCTCGGCCAGGAGTCACAGAAAACAAG	68 °C	Takara PrimeSTAR HS DNA Polymerase
SymkaLHC3R1	TCTCTCGAATTCCGTGTGCTTGTGAAACTTTTATC	68 °C	Takara PrimeSTAR HS DNA Polymerase
DinoSL	NCCGTAGCCATTTTGGCTCAAG	58 °C	Takara PrimeSTAR HS DNA Polymerase
KbrSRP-U6R1	CAGAGATCAAGACATGCTTCAGGAC	58 °C	Takara PrimeSTAR HS DNA Polymerase
gfpNF2	AACTAGTATGGCTAGCAAAGGAGAAGAACTTTTC	5 cycles at 55 °C and 25 at 62 °C	Takara PrimeSTAR HS DNA Polymerase
gfpNR	TATGATCATCATTTGTAGAGCTCATCCATGCCA	5 cycles at 55 °C and 25 at 62 °C	Takara PrimeSTAR HS DNA Polymerase
arr2F	GAGAACTAGTATGGTGAAGGA	57 °C	Takara PrimeSTAR HS DNA Polymerase
arr2R	TCTCTGATCACTAATCCTCG	57 °C	Takara PrimeSTAR HS DNA Polymerase
OxyRhodF2	CACTACTTCMGNATCTTCAACTC	60 °C	Takara PrimeSTAR HS DNA Polymerase
OxyrhodR	CAGAGGMACRGTCARCARCCARTC	60 °C	Takara PrimeSTAR HS DNA Polymerase
Rhod_interspacerF	GAGAACTAGTAATTTTGGGAGTTGGGCT	57 °C	Takara PrimeSTAR HS DNA Polymerase
Illu-DSL	TCGTCGGCAGCGTCAGATGTGTATAAGAGACAGTCCGTAGCCATTTTGGCTCAAG	68 °C	Takara PrimeSTAR HS DNA Polymerase
SymkaLHC3R1	TCTCTCGAATTCCGTGTGCTTGTGAAACTTTTATC	68 °C	Takara PrimeSTAR HS DNA Polymerase
arr2Q1F	TACCACGGAACCAAGGCGAACT	60 °C	SsoAdvanced Universal SYBR Green Supermix
arr2Q1R	CCAAGCCAGACAGCGACATAGC	60 °C	SsoAdvanced Universal SYBR Green Supermix
arr2Q1Fa	GAGATACCACGGAACCAAGGCGAACT	60 °C	SsoAdvanced Universal SYBR Green Supermix
arr2Q1Ra	GAGACCAAGCCAGACAGCGACATAGC	60 °C	SsoAdvanced Universal SYBR Green Supermix
MdT	TCAACGATACGCTACGTAACGTAATACGACTCACTATAGGGTTTTTTTTTTTTTTTTVN	42 °C	Reverse Transcriptase

**Table 3 microorganisms-08-00126-t003:** Adequate pulse code settings for transformation of DinoIII-*gfp*, DinoIII-*arrO*, and DinoIII-*arrO*-N into *Oxyrrhis marina.*

	Average Cell Counts 24 h after Electroporation (cells/well)	Success with *gfp* *	Number of Trials with *arrO*	Number of Trials with *arrO* with Long-Term Survival in Rifampin **	Number of Trials with *arrO*-N	Number of Trials with *arrO-*N with Long-Term Survival in Rifampin **
DS-137	1150	Yes	8	2	2	0
DS-134	2060	No	5	1	2	1
ED-150	3830	Less Bright	2	0	2	0
DS-138	4190	No	6	1	2	2
DS-130	7610	Yes	4	0	2	0
DS-150	7820	No	6	1	2	0
DS-120	16,980	Less Bright	8	1	2	1

***** “Success with *gfp*” is defined as strong green fluorescence 3 months after transformation, as demonstrated in Appendix A. ** Cells survived > 1 month and RNA and/or DNA work was performed.

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
