# Peer review of "Nuclear Gene Transformation in the Dinoflagellate Oxyrrhis marina"

_microorganisms, 2020, doi:10.3390/microorganisms8010126_

Round 1

Reviewer 1 Report

This work is clear, well written, well thought out, rigorous and well implemented. These tools will help a great deal in elucidating the function of many of the unknown genes in this phylum. A delight to review.

Author Response

Point 1: This work is clear, well written, well thought out, rigorous and well implemented. These tools will help a great deal in elucidating the function of many of the unknown genes in this phylum. A delight to review.

Response 1:  We thank the reviewer for the positive comments and tried to address the minor spelling issues we had in the manuscript.

Reviewer 2 Report

Overall I found this an interesting and scientifically sound study outlining a transformation protocol for a dinoflagellate species.

My biggest overall concern is that from the detail in the mthods I could repeat the study. Given this a protocol driven paper this needs to be fixed. For example, The Testing Oxyrrhis marina resistance to selection markers section of the methods contains insufficient information for replication. Please add extra information here and throughout the methods section.

Please do not use the words; basal, early-branching or late-branching, these terms are no longer considered phylogenetically valid. This needs to be corrected in the manuscript.

In some instances, in the manuscript your word choice is a little strange. For example, most exciting on line 75, excel on line 87, respectfully on line 132, italicize species name on line 274. There are many other instances throughout, but I have not made an extensive list.

Author Response

Point 1: My biggest overall concern is that from the detail in the methods I could repeat the study. Given this a protocol driven paper this needs to be fixed. For example, The Testing Oxyrrhis marina resistance to selection markers section of the methods contains insufficient information for replication. Please add extra information here and throughout the methods section.

Response 1: We thank the reviewer for the positive and helpful comments.  We have gone through the methods section and added more information to help make the method more easily replicated in other laboratories.

Point 2: Please do not use the words; basal, early-branching or late-branching, these terms are no longer considered phylogenetically valid. This needs to be corrected in the manuscript.

Response 2: We have also removed most of the areas where we mentioned basal, early- or late-branching, and replaced them with early diverging or early branching in a few places where it is important to indicate the phylogenetic position of O. marina.  To our knowledge, early diverging as well as early branching are still commonly used although "primitive" and "advanced" have gone outdated in the phylogenetic/biological world (see Janouškovec et al., 2017). If the reviewer has alternative terminologies to describe a lineage that branches out from the root earlier than other species, we would appreciate the recommendation to us.

Janouškovec, J., Gavelis, G.S., Burki, F., Dinh, D., Bachvaroff, T.R., Gornik, S.G., Bright, K.J., Imanian, B., Strom, S.L., Delwiche, C.F. and Waller, R.F., 2017. Major transitions in dinoflagellate evolution unveiled by phylotranscriptomics. Proceedings of the National Academy of Sciences114(2), pp.E171-E180.

Point 3: In some instances, in the manuscript your word choice is a little strange. For example, most exciting on line 75, excel on line 87, respectfully on line 132, italicize species name on line 274. There are many other instances throughout, but I have not made an extensive list.

Response 3: We made sure all species and genes are italicized correctly and tried to modify all of our strange word choice throughout the manuscript and hope that this new version reads better.